# The Effect of Functional Motivation on Future Intention to Donate Blood: Moderating Role of the Blood Donor’s Stage

**DOI:** 10.3390/ijerph18179115

**Published:** 2021-08-29

**Authors:** Lifan Zhang, Meilin Yao, Hongrui Liu, Shuang Zheng

**Affiliations:** Faculty of Psychology, Beijing Normal University, Beijing 100875, China; zhanglifan@mail.bnu.edu.cn (L.Z.); hrliu@bnu.edu.cn (H.L.); zhengshuang@mail.bnu.edu.cn (S.Z.)

**Keywords:** blood donation, intention, function approach, motivation, blood donor stage

## Abstract

The low retention rate and lack of long-term blood donors are still worldwide problems, and the fostering and retaining of more blood donors has posed an urgent problem that requires a solution. The purpose of this study was to explore the role of functional motivation and blood donor stage in the future intention to donate blood. Three sites in China provided 252 blood donors who filled out a questionnaire including the volunteer function inventory, their future intention to donate blood, number of blood donations and demographic information. Based on a functional motivation approach, we investigated the motivations that influence unpaid blood donors’ future intention through qualitative interview methods. The results showed that blood donor stage moderated the relationship between values motivation and future intention to donate blood. More specifically, the predictive effect of values motivation on future intention decreased gradually from new donors to regular donors. Our study found that the effect of various motivations on blood donors’ intention was conditional; this effect changed with the development of blood donor behavioral stages. The results are significant in practice for establishing effective intervene strategies in maintaining blood donors.

## 1. Introduction

The number of blood donors in China has increased from 4 million to 15 million since 1998, when the promulgation of the Law of the People’s Republic of China Blood Donation was enacted; however, a low retention rate and a lack of long-term donors still pose problems. World Health Organization (WHO) suggested that, to meet a country’s blood need sufficiently, the voluntary unpaid blood donation rate should be 10‰ to 30‰ [1]. Although the Chinese blood donation rate reached 11‰ in 2019 (up to the minimum standard [2]), it remains relatively low compared to the national average level [3,4]. Because China is still one of the countries with a blood supply shortage, fostering and retaining more blood donors have become an urgent problem requiring a solution.

Multiple perspectives have been taken to discuss people’s blood donation intention. Quite a few previous studies on the intention to donate blood were based on the theory of planned behavior (TPB) and its extensions, which indicated that a logical sequence of cognition process caused future blood donation intention (e.g., attitude, subjective norms, and perceived control) [5,6,7,8,9]; some other researchers focused on the demographic characteristic aspects such as age, gender, education, and experience [10,11,12]. Furthermore, as a critical factor influencing blood donation intention, motivation has been studied by scholars, but most of these studies were based on the grounded theory and considered motivation and its effect as static [13,14,15,16]. A few pieces of research have discussed individuals’ blood donation intention using a systematic motivational theory [17,18,19], especially in Chinese culture. In volunteerism, many studies have proved the predictive effect of functional motivations on volunteer intention [18,20,21,22]. According to the functional motivation approach, people seek to fulfill various needs when they participate in volunteer activities, and people activated by different psychological functions may show the same behavior [20]. As a form of volunteering, people who donate their blood may be promoted by different motivations. Exploring the future blood donation intention in terms of the functional motivation approach therefore provides insight into how to inspire more people to participate and persist in blood donation. Meanwhile, since motivation and its effect could change at different times for the same person, more and more researchers have considered blood donation as developmental, and the motivational dimensions that sustain it also change during the donor’s career life [17,23,24], so we would take blood donors’ stage into account. To address gaps in the literature, we investigated the relationship of various motivations with the future intention to donate blood, and thus considering the role of blood donor stage is necessary.

### 1.1. Motivation to Donate Blood

People’s motivation to donate blood has been the focus of many researchers. Previous studies on the characteristics of blood donation motivation in various blood donors showed that (a) donors with higher levels of education were more likely to donate because “it was the right thing to do” than because it has health benefits, (b) what motivated first or new donors was related primarily to the influence of friends and setting [13,25], and (c) women were more likely than men to donate for altruistic and humanitarian reasons [25,26]. Researchers have held various ideas about people’s motivation to donate blood, one of which is that blood donation is an altruistic, other-, or social-aimed behavior motivated by humanism [27]. Another derives from the benevolence hypothesis, emphasizing that donors most value their own feelings [28,29].

Most researchers, however, held that blood donors’ motivations are diverse. Some researchers listed motivations to donate blood including altruism, health improvement, peer pressure, and rewards [15,16,30]. The results of a meta-analysis suggested the following eight motivations for blood donors: the convenience of collection site, prosocial motivation, personal values, perceived need for donation, indirect reciprocity, marketing communications, incentives, and social norms [31]. The major of previous research about blood donation motivation was mainly based on the grounded theory with a bottom-up approach [13,14,15,16] and the motivation of blood donor’s was seen as relatively constant and single. With the growth of the research on blood donor behavior, more and more researchers considered blood donation as a process, since blood donors’ motivation could change in a different donor career life [17,23,24,32]. Further investigation on blood donor motivation with a systematic and developmental perspective is still needed.

The functional motivation approach is one of the most accredited approaches for study the underlying motivations of volunteering. From the functional perspective, people volunteer to fulfill various psychological needs; their behavior in volunteering may seem similar, but internal motivation varies [20,33,34]. Motives serve various psychological functions and social needs, and they are inseparably related to the generation and maintenance of volunteerism [20]. The volunteer function inventory (VFI), derived from the functional approach, includes six motives for volunteering [20]: (a) values (to express altruistic and humanitarian values), which may be the key feature distinguishing volunteers from non-volunteers [35]; (b) understanding (to acquire new experiences and skills, learning in practice); (c) social (to make new friends or volunteer as others expect, reflects relationships with others); (d) career (to obtain experience related to the current career or a future career); (e) protective (to reduce negative affect from a self-defense perspective); and (f) enhancement (to pursue positive feelings about oneself, such as self-esteem and sense of value). The functional approach emphasizes the role that the volunteer’s need for satisfaction played throughout the volunteer activity [36]. Similarly, with blood donation as a form of volunteering, people could be promoted by psychological functions to donate their blood. Few researchers have taken the functional motivation approach into account despite the important role that it has played in understanding blood donor’s motivation [17,19,24,37]. Thus, investigating the influencing factors in blood donation using the functional motivation approach may shed light on how a volunteer’s motivation affects intention as well as the behavior of blood donation.

### 1.2. Motivation and Future Intention to Donate Blood

Most previous studies about future intention to donate blood were mainly based on the TPB [5,6,7,8,38,39], which suggested that attitude, subjective norms, and perceived behavioral control are the main predictors of intention [9]. Although TPB provides a reasonable framework to explain the intention underlying behavior, it emphasized the primary stage of behavior activation and involved little info related to behavior change or sustenance. The volunteer functional approach, which emphasized the matching of individuals’ motivations and activities to make volunteering more lasting, provides a new perspective into the sustaining of blood donors’ behavior. A few studies have shown that functional motivations were important in predicting individuals’ future blood donation intention [18,21,37,40]. The value motivation, one of the functional motivations, was also found to play a key role in predicting individuals’ intention to volunteer [18]. Although the functional approach has been applied widely as the theoretical basis for studies about volunteering, it has rarely been investigated in the field of blood donation. In particular, most studies on the motivation to donate blood have involved donor demographics but have failed to reveal their influencing mechanism [17,23,25]; therefore, exploring the effects of motivation in the context of the function approach can be meaningful.

### 1.3. The Moderating Role of the Stage Model of Blood Donation

The stage model of blood donation was proposed [32,41] based on the transtheoretical model (TTM), which explained the change in behavioral intention [42]. Ferguson proposed a five-stage model of blood donation as follows: (a) precontemplative stage (nondonors who do not think about blood donation in any situation), (b) contemplative stage (nondonors who are aware of the need to donate), (c) preparation stage (potential donors who prepare to act), (d) action stage (first-time donors or committed donors), and (e) maintenance stage (committed repeat donors) [41]. Having examined the stage model of blood donors’ behavior qualitatively and quantitatively, Ferguson and Chandler then proposed three stages in blood donation: precontemplative stage, intention–preparation stage and action–maintenance stage [32]. Recently, researchers further classified the blood donor stages based on the stage model and practice by the number of donations: 1 to 4 are new donors, 5 to 15 are loyal donors, and 16 or more are regular donors [17]. This standard was adopted in our study to describe the types of donors. In the TPB blood donation, behavior was regarded as continuing, but in the stage model of blood donation, donors’ behaviors do not always transition from one stage to another in sequence. Instead, blood donation behaviors are considered to fall into stages. According to the stage model of blood donation, donors’ behavior varies with the stages. Taking a donor’s stage into account may be beneficial to understanding blood donation behavior more accurately and contribute to a plan to encourage more people to engage in blood donation.

From the functional motivation perspective, individuals’ motivations vary with volunteer activities [20]. Previous studies have shown that with the development of behavior, blood donors’ motivations changed from non-donors to regular donors, and individuals’ motivation transformed from amotivation to intrinsic motivation, such as helping others and personal values [17,25]. Wong and Foo mentioned that functional motivations varied when volunteering experiences were considered, such as the values and social functions increased with people’s volunteering experience and the career function was less important for volunteers with more experience, and according to these variations, researchers could posit that volunteers were motivated by particular functions at various levels of experience [43]. The stage of blood donation is an important indicator of blood donors’ experience; therefore, the following assumption underlies the current research: the effect of motivation on intention changes with blood donors’ stage.

### 1.4. The Current Study

To encourage more people to become regular voluntary blood donors, exploring the factors influencing the intention to donate blood is necessary. Previous researchers paid attention primarily to the blood donor motivation with a bottom-up approach [13,14,15,16], and blood donor motivation was understood as relatively constant and single. Further studies are still needed to investigate the relationship between functional motivation and future intention from a systematic and developmental perspective, since there aren’t many similar studies regarding Chinese blood donors. Does functional motivation affect the donor’s intention? For donors in the various stages, do certain motives dominate in their intention to donate in the future? Such questions require answers, which will provide useful information on how to encourage and sustain more blood donors in China. Thus, the purpose of the current study was to investigate the effects of a variety of functional motivations on the intention to donate blood in the future while considering the role of the donation stage.

## 2. Materials and Methods

This cross-sectional study was conducted using two previously validated scales: an adapted version of the volunteer function inventory [17,19] and future intention to donate blood [39]. Blood donor stage was defined by the number of previous blood donations [17,32] and participants’ demographic information was collected before the questionnaire. This study was approved by the Ethics Committee of the School of Psychology, Beijing Normal University.

### 2.1. Participants and Procedure

The sample size was estimated using the Qualtrics sample size calculator [44]. For an average blood donor population of above 15,000 every year in Yuxi, China (data from the Department of Statistic of Yuxi Central Blood Station), the minimum number of estimated samples was 266 with a 90% confidence interval and 5% margin of error; therefore, we recruited a total 288 blood donors in this study. Thirty-six of these participants were excluded from the study because they left the program; therefore, 252 donors participated in the current study (44.4% female, Mage = 37.50, SD = 10.15). According to the classification of previous researchers [17,19], 89 were new donors (having donated blood 1–4 times); 87 were loyal donors (5–15 donations); and 76 were regular donors (more than 16 donations).

Blood donors were enrolled from three blood collection sites. All donors signed consent forms and completed the questionnaire anonymously at the donation station. Souvenirs were given when they finished the questionnaire.

### 2.2. Measurements

Motivation to donate blood was assessed with the VFI [20]. Career motivation on the VFI was not applicable in the blood donation context, so the adapted version of VFI without the dimension of career development was used [17,19]. The questionnaire was comprised of 25 items, which participants rated on a 7-point scale ranging from 1 (strongly disagree) to 7 (strongly agree). Sample items covering the five motivations included the following: (a) values: “I feel it is important to help others by donating blood”; (b) understanding: “Blood donation allows me to gain a new perspective on things”; (c) social: “Others with whom I am close place a high value on blood donation”; (d) protective: “No matter how bad I’ve been feeling, blood donation helps me to forget about it”; and (e) enhancement: “Blood donation makes me feel needed”; *α* = 0.76 to 0.84. For the first application of the VFI in Chinese blood donors, we conducted a confirmed factor analysis (KMO = 0.91 and Bartlett’s test of sphericity = 3202.22, *p* < 0.001), and these five factors noted above explained 63.19% of total variance beyond the critical value. Given the results, the questionnaire was acceptable.

Future intention to donate blood was measured with three items [5] rated on a 7-point scale, ranging from 1 (strongly disagree) to 7 (strongly agree). A sample item was “I’d like to donate blood again if I could” (*α* = 0.91).

The number of previous blood donations was used as the indicator of blood donor stage [17,19]. Blood donors who donated 1 to 4 times were new donors, those who donated 5 to 15 times were loyal donors, and those who donated 16 times or more were regular donors.

Demographic variables in the current study included age, gender (coded as a dummy variable with male = 1, female = 0), and socioeconomic status (SES), which included occupation, annual income, and educational level.

### 2.3. Data Analysis

The statistical analyses included reliability analyses, descriptive analyses, and analysis of variances (ANOVAs), and the moderating effects were conducted with the software SPSS Statistics 23.0.

## 3. Results

### 3.1. Descriptive Statistics

Main variables’ means, standard deviations, and correlations analysis appear in Table 1. Among the five functional motivations, the values motivation was the highest (M = 5.86, SD = 0.99), then social (M = 5.29, SD = 1.17), enhancement (M = 4.93, SD = 1.29) and understanding (M = 4.43, SD = 1.44) motivations; the protective motivation was the lowest (M = 3.34, SD = 1.28). Participants’ average number of blood donations was 9.94 (SD = 8.87). All of the motivations positively correlated with the intention to donate blood except for protective motivation (which ranged from 0.25 to 0.49). The number of blood donation was positively correlated with understanding, enhancement, and protective motivations (which ranged from 0.14 to 0.24) but were unrelated to social and values motivations. In addition, we conducted a series of ANOVAs to explore the different motivations between stages further. The results showed that the main effects of understanding, enhancement, and social motivations were significant; *F*s (2, 251) ranged from 6.82 to 7.78 and *p*s < 0.01, *η*^2^s ranged from 0.05 to 0.06. A post hoc Bonferroni test indicated that loyal and regular donors reported higher understanding, enhancement, and social motivations than new donors (*p*s < 0.01), but there was no difference between loyal and regular donors.

### 3.2. Moderation Analyses

To examine the moderating role of blood donor stage in the relationship between motivation and future blood donation intention, we conducted a three-step hierarchical regression analysis. All control variables and independent variables were standardized. The first step was to designate gender, age, and SES as control variables, the second step was to enter the variables of motivations and blood donor stages, and the third step was to enter interaction terms of motivations × blood donor stage as predictors. This procedure was performed two times because blood donor stage was a categorical variable with three levels. This categorical variable was coded as two dummy variables twice. In the first hierarchical regression, we treated new donors as a control variable for comparing with loyal donors and regular donors. Specifically, new donors were denoted as λ_1_ = 0, λ_2_ = 0, loyal donors were denoted as λ_1_ = 1, λ_2_ = 0, and regular donors were denoted as λ_1_ = 0, λ_2_ = 1. In the second hierarchical regression, loyal donors served as a control variable for further examining their difference from regular donors. Specifically, new donors were denoted as λ_3_ = 1, λ_4_ = 0, loyal donors were denoted as λ_3_ = 0, λ_4_ = 0, and regular donors were denoted as λ_3_ = 0, λ_4_ = 1.

The results, summarized in Table 2, showed that age, gender, and SES were not significantly related to the intention to donate blood (Step 1). Overall, the impact of protective (*β* = −0.16, *p* < 0.05), social (*β* = 0.21, *p* < 0.01), and values (*β* = 0.44, *p* < 0.001) motivations on intention was significant (Step 2). The regression analysis in Step 3 showed that blood donor stage moderated the relationship between values motivation and intention to donate blood significantly. Specifically, a significant difference emerged between new donors and regular donors in terms of the effect of values motivation on the intention to donate (*β*λ_2_ = −0.52, *p* < 0.01). Additionally, this effect was only marginally significant between new donors and loyal donors (*β*λ_1_ = −0.28, *p* = 0.06) but was not significant between loyal donors and regular donors (*β*λ_4_ = −0.25, *p* = 0.16). Moreover, no moderation effect of blood donor stage was shown in the relationship between the intention to donate blood and the understanding, enhancement, protective, and social motivations.

To reveal the moderating role of blood donor stage in the relationship between values motivation and the intention to donate blood, simple slope tests were used. Based on a pick-a-point approach [45], the interaction diagram was produced with one-unit standard deviations below and above the mean value (see Figure 1). For the intention to donate blood, the predictive effect of values motivation was significant for new donors (simple slope = 0.70, t = 6.95, *p* < 0.001) and loyal donors (simple slope = 0.42, t = 3.92, *p* < 0.001), but it was not found to have any significant effect for regular donors (simple slope = 0.18, t = 1.30, *p* = 0.20).

## 4. Discussion

The issue of inspiring and retaining blood donors is always a concern of researchers and medical workers. The current study contributes to the knowledge of blood donors’ behaviors by revealing the effect of motivations on future intention to donate blood, considering the blood donor stages. Our results of ANOVAs showed that loyal and regular donors reported higher understanding, enhancement and social motivations than new donors, but there was no difference between loyal and regular donors. For the social motivation, this finding is in line with the study of Guiddi et al. [17], which showed that social motivation increase with an increase in donation number. It could be that donors can develop new friendships in the context of blood donation organizations, especially since most blood donation centers in China will host social events and give rewards for donors, as these donors would have more time and chance to communicate with other donors and medical workers. However, the findings of understanding and enhancement motivations were inconsistent with those of two previous studies [17,46], which also drew different conclusions. One of the possible explanations for these results might be that with the development of donors’ career life, loyal and regular donors learned more about blood donation, they gained knowledge and achieved the promotion of self-acceptance and self-esteem compared with new donors. However, considering the limited number of studies about blood donors’ functional motivations [17,46], we should be cautious about concluding, as further research in different cultures and populations is still needed.

In accord with Saha and Chandra’s study [40], our results showed that protective motivation negatively predicted the future intention to donate blood. In volunteerism, the protective motivation was related to individuals’ psychological process of ego defense, such as reducing bad moods [20]; however, compared with other volunteer activities, blood donation involves more health-related risk and negative emotions, such as hypoglycemia, fainting, pain of piercing, and fear [6,47]. Accordingly, if people have a strong will to protect themselves from negative feelings, they may choose other volunteer activities instead of persisting in blood donation. Moreover, in line with the previous study [40], social motivation was found to positively predict the intention to donate blood. A likely explanation is that peer recruitment was an important way to attract new donors [48], and in this study, social motivation reflected the social process of individuals’ influence on one another during blood donation, such as the expectations of family and encouragement from friends, which could impact blood donors’ intention.

The current study showed that values motivation positively predicted the intention to donate blood, which was consistent with previous studies [18,40]. Similarly, this result was in accordance with several existing studies in volunteerism, which showed that values was identified as an important factor to the subject’s willingness to volunteer [49,50,51]. Values motivation reflected the values related to altruistic and humanitarian concern for others [20], and donors with higher-level values motivation were more likely to donate in the future; this effect varied with blood donors’ stages. Specifically, with a change of blood donation stage, the effect of values motivation on future intention to donate showed a decreasing trend; this result is consistent with a previous qualitative study we conducted [52]. Controlling the variables of blood donors’ age, gender, and SES, the motivation of helping others and saving lives played key roles in the future intention to donate for both new donors and loyal donors. On the one hand, the decreasing impact of values motivation on intention provided evidence of the stages of blood donation, for the blood donor behavior changed with their stages [32]. On the other hand, it also indicated that donors’ behaviors might become habituated because of the internalization of their values motivation. It is worth mentioning that, differently from previous studies [17,37,40], our research further uncovered the effect of values motivation on blood donation intention with the change of stages. Furthermore, some other factors may influence regular donors; for example, the convenience of accessing the blood collection sites may be one of the major factors for regular donors to donate again [15,53]. This study also updated the knowledge that most regular donors treat blood donation as a natural part of their daily lives, and they would continue to donate blood for the foreseeable future without incentives.

The findings of this study are important for developing strategies to maintain blood donors in practice. To impel donors to give blood in the long run, not only could blood center personnel encourage donors in various ways by considering the types of blood donors, but family and peers could also play important roles in blood donor recruitment. Although the researchers’ study took into consideration the effect of functional motivations on the intention to donate blood with a cross-sectional method, future longitudinal studies would improve the ecological validity of the study. In addition, some studies have revealed that factors like the internalization of motivations and role identity may play key roles during the upgrade of blood donation stages [25]; therefore, further efforts on the internalization of motivation and the development of role identity under the stage model of blood donor behavior would be valuable.

## 5. Conclusions

The current study provided insight to possible implications for blood donors’ behaviors by examining the effect of motivations on future intention to donate blood with the moderation of the blood donor stages. Based on our findings, firstly, compared with other functional motivations, value, social, and protective motivations were the key predictors of blood donors’ future intention, as these effects indicated that blood donors who with more altruistic belief. Those looking for more social support and less ego-defense were more willing to choose to donate blood in the future; this could be an inspiration for recruiting more blood donors in practice. Secondly, the three-step hierarchical regression analysis showed that there was a moderation effect of blood donor stage in the relationship between value motivation and the intention to donate blood, the predictive effect of values motivation was significant for new donors and loyal donors, but not for regular donors, which implied that donors’ behaviors may internalize values motivation. Finally, our findings confirmed that the blood donors’ behaviors changed with their blood donor stage. Researchers, medical workers, and managers should pay more attention to blood donors’ experience for the maintenance of the blood donor group.

## Figures and Tables

**Figure 1 ijerph-18-09115-f001:**
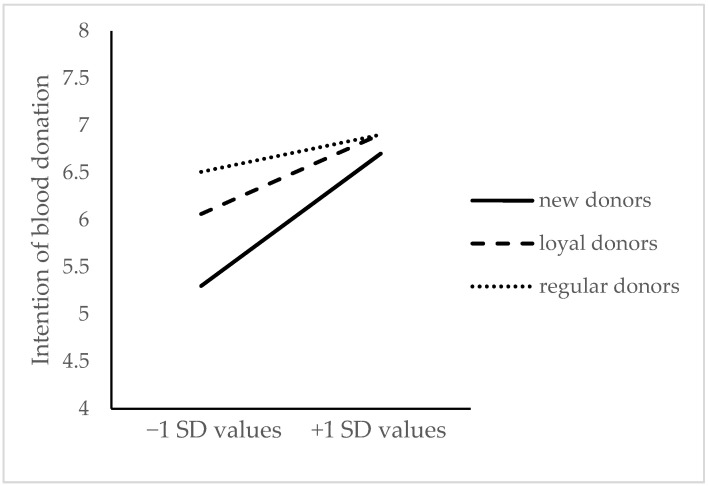
Intention to donate blood explained by moderation of understanding motivation and blood donor stage.

**Table 1 ijerph-18-09115-t001:** Means, standard deviations, and correlations coefficients of the main variables ^1^.

Variables	1	2	3	4	5	6	7	8	9	10
1 Gender	-									
2 Age	0.07	-								
3 SES	−0.12	−0.28 **	-							
4 Understanding	0.12	0.21 **	−0.15 *	-						
5 Enhancement	0.05	0.15 *	−0.16 *	0.78 **	-					
6 Protective	0.00	0.00	−0.05	0.66 **	0.66 **	-				
7 Social	0.00	0.22 **	−0.08	0.42 **	0.44 **	0.25 **	-			
8 Values	0.05	0.12	−0.07	0.52 *	0.49 **	0.25 **	0.42 **	-		
9 Number of blood donations	0.12 *	0.46 **	−0.31 **	0.24 **	0.20 *	0.14 *	0.10	0.03	-	
10 Intention to donate blood	0.04	0.27 **	−0.19 **	0.25 **	0.30 **	0.04	0.41 **	0.49 **	0.28 **	-
M	0.56	37.50	0.00	4.43	4.93	3.34	5.29	5.86	9.94	6.28
SD	0.50	10.15	0.78	1.44	1.29	1.28	1.17	0.99	8.87	1.04

^1^ Note. * *p* < 0.05, ** *p* < 0.01. SES = Socioeconomic Status.

**Table 2 ijerph-18-09115-t002:** Results of hierarchical regression analyses on future intention of blood donation ^2^.

Variables	Dummy Variable A (λ_1_, λ_2_) *β*			Dummy Variable B (λ_3_, λ_4_) *β*		
	Step 1	Step 2	Step 3	Step 1	Step 2	Step 3
Gender	0.02	0.01	0.11	0.02	0.01	0.11
Age	0.02	0.00	−0.00	0.02	0.00	−0.00
SES	−0.16	−0.07	−0.06	−0.16	−0.07	−0.06
Understanding		−0.14	−0.06		−0.14	−0.20
Enhancement		0.14	0.26		0.14	0.28
Protective		−0.16 *	−0.26		−0.16 *	−0.28 *
Social		0.21 **	0.22 *		0.21 **	0.22 *
Values		0.44 ***	0.70 ***		0.44 ***	0.42 ***
λ_1_/λ_3_		0.52 ***	0.48 ***		−0.52 ***	−0.48 ***
λ_2_/λ_4_		0.72 ***	0.71 ***		0.20	0.22
Understanding × λ_1_/λ_3_			−0.14			0.14
Understanding × λ_2_/λ_4_			−0.06			0.08
Enhancement × λ_1_/λ_3_			0.03			−0.03
Enhancement × λ_2_/λ_4_			−0.33			−0.36
Protective × λ_1_/λ_3_			−0.01			0.01
Protective × λ_2_/λ_4_			0.29			0.31
Social × λ_1_/λ_3_			−0.01			0.01
Social × λ_2_/λ_4_			−0.03			−0.03
Values × λ_1_/λ_3_			−0.28			0.28
Values × λ_2_/λ_4_			−0.52 **			−0.25
*R* ^2^	0.09	0.41	0.48	0.09	0.41	0.48
Δ*R*^2^	0.09	0.32	0.07	0.09	0.32	0.07
Δ*F*	7.63 ***	18.90 ***	3.37 ***	7.63 ***	18.90 ***	3.37 ***

^2^ Note. * *p* < 0.05, ** *p* < 0.01, *** *p* < 0.001.

## Data Availability

The data are not publicly available to protect the confidentiality of the participants.

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
