# Peer review of "The Effect of Functional Motivation on Future Intention to Donate Blood: Moderating Role of the Blood Donor’s Stage"

_ijerph, 2021, doi:10.3390/ijerph18179115_

Round 1

Reviewer 1 Report

The topic of the manuscript is interesting but the text by itself is consisting mainly of the description and analysis of questionnaire results. Stronger and more convincing argumentation for the conduction of presented research should be provided. 

What are the novel aspects of the presented research? It would be very supportive to add the comparison of the results with results of other similar studies.

It would be convenient to provide more precise information about the blood donation rate of China compared to other countries (this aspect is shortly mentioned in the Introduction section).

Please reference at least some of the studies described in lines 34-39. How many sources were reviewed?

The short introductory sentence before the introduction subchapters could be supportive for better orientation of the reader.

Please introduce research questions presented in lines 138 and 139.

Line 137: „...especially for China donors.“ Please provide resonation why this problematics should be particularly studied in China. A more detailed explanation of this aspect that was already mentioned in the first part of the Introduction should be provided.  It is closely related to the first comment.

How were survey participants selected? What confidence level and confidence interval were used for defining the sample of respondents?

The Materials and Methods section starts very abruptly. A paragraph is needed to introduce the reader to the methodological framework of the presented research.

It would be convenient to specify the difference between blood donor categories that are used in the research sample description – line 149. This aspect is explained too late (in lines 174-175). Who and on the basis of what decided the ranges of these categories?

Please delete the text provided in lines 180-182. Instead of it use an introductory sentence or paragraph for defining the structure of this section.

Use capital letter in the expression “10 intention to donate blood” and explain the abbreviation SES in Table 1.

In the first part of your work, you mention the TPB model. Did you considered also other models related to the topic of your study (TRA, PMT, HBM, eg.)

Is the motivation to donate blood somehow influenced by the blood group of the donor?

A major upgrade and careful revision and rewriting are needed in terms of structure, presentation of methodology and results, and conclusions.

Reviewer 2 Report

The authors have done a very good job in presenting a clear rationale and research design for the paper. More attention is needed in the Introduction to show how the functional motivation approach extends and aligns to existing research on blood donation motivations (i.e. where are the overlaps and which functions offer new insight). This will also help in the discussion, to make the contribution stronger.

There is one paper of note, that is currently missing that applies the VFI to a blood donation context: Alfieri, S., Pozzi, M., & Pistoni, C. (2020). The blood donation function inventory: Adaptation of the voluntary function inventory for a psychological approach to blood donors. Journal of Civil Society16(1), 61-76.

It seems there is instructional text under the head Results (page 4); "This section may be divided by subheadings. It should provide a concise and precise description of the experimental results, their interpretation, as well as the experimental conclusions that can be drawn."

The moderation effect of the values function could be linked more with the literature, as research suggests the more we donate, our motivations shift from external to internal. Yet, your results show that the effect of 'values' on intention to donate is stronger for newer donors. Is this consistent with results found in a volunteering context? To further understand motivational differences between donor career stage, it would be good to include ANOVAs in addition to the moderation effects. 
